# A Comparative Study of Canine Mesenchymal Stem Cells Isolated from Different Sources

**DOI:** 10.3390/ani12121502

**Published:** 2022-06-09

**Authors:** Filip Humenik, Marcela Maloveska, Nikola Hudakova, Patricia Petrouskova, Lubica Hornakova, Michal Domaniza, Dagmar Mudronova, Simona Bodnarova, Dasa Cizkova

**Affiliations:** 1Centre of Experimental and Clinical Regenerative Medicine, The University of Veterinary Medicine and Pharmacy in Kosice, 041 81 Kosice, Slovakia; filip.humenik@uvlf.sk (F.H.); marcela.maloveska@uvlf.sk (M.M.); nikola.hudakova@student.uvlf.sk (N.H.); patricia.petrouskova@uvlf.sk (P.P.); 2University Veterinary Hospital, The University of Veterinary Medicine and Pharmacy in Kosice, 041 81 Kosice, Slovakia; lubica.hornakova@uvlf.sk (L.H.); michal.domaniza@student.uvlf.sk (M.D.); 3Institute of Microbiology and Immunology, The University of Veterinary Medicine and Pharmacy in Kosice, 041 81 Kosice, Slovakia; dagmar.mudronova@gmail.com; 4Department of Pneumology a Phtiseology, Faculty of Medicine, University of Pavol Jozef Safarik, 041 80 Kosice, Slovakia; simona.bodnarova@upjs.sk; 5Institute of Neuroimmunology, Slovak Academy of Sciences, 845 10 Bratislava, Slovakia

**Keywords:** canine mesenchymal stem cells, morphology, phenotype, multilineage potential

## Abstract

**Simple Summary:**

The present study describes differences in the isolation yield, morphology, presence of surface markers and proliferation capacity but not in the multilineage potential of canine MSCs isolated from bone marrow, adipose tissue and amnion. Among all the MSCs analysed, AT-MSCs showed the highest isolation yield, phenotype homogeneity, proliferation capacity and osteogenic and chondrogenic potential. In addition, for BM-MSCs and AM-MSCs, we uncovered some differences that need to be considered during isolation, expansion and phenotyping prior to their possible application in targeted regenerative veterinary medicine.

**Abstract:**

In this study, we provide comprehensive analyses of mesenchymal stem cells (MSCs) isolated from three types of canine tissues: bone marrow (BM-MSCs), adipose tissue (AT-MSCs) and amniotic tissue (AM-MSCs). We compare their morphology, phenotype, multilineage potential and proliferation activity. The BM-MSCs and AM-MSCs showed fibroblast-like shapes against the spindle shape of the AT-MSCs. All populations showed strong osteogenic and chondrogenic potential. However, we observed phenotypic differences. The BM-MSCs and AT-MSCs revealed high expression of CD29, CD44, CD90 and CD105 positivity compared to the AM-MSCs, which showed reduced expression of all the analysed CD markers. Similarly, the isolation yield and proliferation varied depending on the source. The highest isolation yield and proliferation were detected in the population of AT-MSCs, while the AM-MSCs showed a high yield of cells, but the lowest proliferation activity, in contrast to the BM-MSCs which had the lowest isolation yield. Thus, the present data provide assumptions for obtaining a homogeneous MSC derived from all three canine tissues for possible applications in veterinary regenerative medicine, while the origin of isolated MSCs must always be taken into account.

## 1. Introduction

Mesenchymal stem cells (MSCs) could be described as unspecialised cells with multilineage differentiation potential and self-renewal capacity [1]. In general, they are located in discrete microenvironments, termed perivascular niches, where they play a key role in maintaining tissue homeostasis and healing processes [2,3,4]. The main sources of mesenchymal stem cells are bone marrow and adipose tissue [5], but these cells can also be obtained from skin, dental pulp, liver, ovarian epithelium, umbilical cord, placenta, amniotic fluid and others [6,7]. Mesenchymal stem cells can differentiate into cells of mesodermal origin, such as chondrocytes, osteoblasts and adipocytes. Several studies also point to the possibility of differentiating into ectodermal or endodermal lineages, such as nerve tissue cells or hepatocytes [1]. However, the transdifferentiation of MSCs from the original mesodermal line into ectodermal cells remains unclear and controversial [8]. Due to the fact of their multilineage differentiating ability, mesenchymal stem cells expand the possibilities of regenerative medicine [9], where their application helps to replace bone tissue and cartilage [5,10]. However, the low survival and transient retention of transplanted MSCs in host tissue [11] indicate that MSCs exert their therapeutic effects via secretion of bioactive factors that provide a favourable microenvironment to facilitate the repair and regeneration of injured tissues [12]. The paracrine effect of MSCs plays an important role, as angiogenesis, neuroprotection, and immunoregulation in the target tissue are affected through the production of important growth and trophic factors or bioactive molecules [13,14]. Due to the lack of understanding of the complexity of secreted bioactive factors, MSC secretome-based therapy in human and veterinary medicine has not yet been fully established [15,16]. One of the obstacles is the variability of the MSC secretome, which is influenced by the donor, tissue source, culture conditions and passage [15,17]. A better understanding of the factors of the biological function that makes up the secretome in relation to its tissue source will allow for the development of more effective and diseases targeted therapy [18]. 

In principle, according to the minimum criteria required for defining MSCs [19], it is necessary to select the appropriate source of biologically active MSCs capable of releasing the specific factors needed for in situ tissue regeneration [20]. We can currently draw on the amount of information available on MSCs isolated from various tissues and their use for alternative treatments in veterinary medicine [21]. Therefore, in the present study, we compared MSCs isolated from three different canine tissues (i.e., bone marrow, adipose tissue, and neonatal amniotic membrane), which have certain advantages/disadvantages in terms of isolation and yields and their biological activities eligible for the regeneration of specific tissues. Our results are consistent with other works [22,23,24] and confirm the ability to isolate MSCs from various canine adults as well as perinatal tissues.

## 2. Materials and Methods

The study was performed after obtaining informed consent from the owners and the approval of the Ethics Committee at the University of Veterinary Medicine and Pharmacy in Kosice on 2 September 2021 (EKVP/2021-01).

### 2.1. Isolation of MSC from Bone Marrow

The bone marrow was harvested from purebred healthy dogs (*n* = 3). Donor 1—male, German Shepherd, 35 kg, 3 years old. Donor 2—female, Cane Corso, 70 kg, 4 years old. Donor 3—male, German Shorthaired Pointer, 32 kg. Before bone marrow collection, all donors were examined (clinical examination, biochemical, and haematological parameters were evaluated). Harvesting was performed under general anaesthesia. The place of the collection was prepared according to all principles of sterility and asepsis as during surgery. We selected the proximal part of the humerus as the sampling site. The collections were performed with an 18 G injection needle and a 10 mL syringe. After penetrating the bone marrow cavity, the bone marrow was aspirated with a 10 mL syringe containing 3 mL of flushing medium. Flushing medium composition: Dulbecco’s modified Eagle’s medium/Nutrient Mixture F12 (DMEM-F12), 10% foetal bovine serum (FBS), 1% antibiotic–antimycotic (ATB+ATM; penicillin–streptomycin–amphotericin B) and 1% glutamine (all Biowest). The aspirate was then centrifuged twice for 10 min at 400× *g*. Cells were plated in cultivation flask T25, at a seeding density of 10^6^ cells/flask, and were cultured in media containing DMEM-F12 + 10% FBS + 2% ATB + ATM at 37 °C and 5% CO_2_. After 48 h of incubation, nonadherent cells were removed. 

### 2.2. Isolation of MSCs from Adipose Tissue

Adipose tissue was harvested from purebred healthy dogs (*n* = 3) and was collected under general anaesthesia, maintaining the principles of sterility and asepsis, from the subcutaneous tissue in the scapular area. Adipose tissue was collected from identical donors as for bone marrow (Section 2.1). We isolated 5–7 g of adipose tissue from each donor. The tissue was then washed with phosphate-buffered saline (PBS; Biowest) containing 2% ATB + ATM, then mechanically dissociated and enzymatically digested with 0.05% collagenase type I and IV (Sigma, Burlington, MA, USA) at 37 °C for 1 h. At the end of the incubation period, the digested tissue was filtered (through a 100 μm cell strainer) to remove tissue fragments, centrifuged at 400× *g* for 10 min, and the obtained stromal vascular fraction (SVF) pellet was resuspended in culture medium consisting of DMEM-F12 containing 10% FBS and 2% ATB + ATM and plated in a T25 tissue culture flask at a concentration of 10^6^ cells/flask and incubated in cultivation medium (DMEM-F12 + 10% FBS + 2% ATB + ATM) at 37 °C and 5% CO_2_. After 48 h, non-adherent cells were removed and, subsequently, the medium was changed twice a week.

### 2.3. Isolation of MSCs from Amniotic Tissue

Amniotic tissues were obtained during caesarean section (new-born puppies, *n* = 6, 62nd day of pregnancy) under strictly sterile conditions. Donors—German rottweiler, weight of puppies approximately 350 g, sex ratio—3:3. The amnions were then washed with PBS containing 2% ATB + ATM; then, the tissue was mechanically dissociated and enzymatically digested using 0.05% collagenase type I at 37 °C for 30 min. At the end of the incubation period, the digested tissue was filtered (through a 100 μm cell strainer) to remove tissue fragments, and the obtained fraction was centrifuged at 400× *g* for 10 min. The obtained pellet was resuspended in DMEM-F12 culture medium + 10% FBS + 2% ATB + ATM. Cells were plated in a T25 culture flask at a concentration of 10^6^ cells/mL and incubated in culture medium (DMEM-F12) + 10% FBS + 2% ATB + ATM at 37 °C and 5% CO_2_ Nonadherent cells were removed, and the medium was subsequently changed twice a week.

### 2.4. Passaging Cells

When the cultivated cell population reached a confluence of approximately 75–80%, we proceeded to passage. To separate the cells from the surface of the culture flask, we used an enzymatic trypsinization method using 0.25% Trypsin EDTA (Biowest), which acted on the cells depending on the level of confluence for 5–7 min at 37 °C. To inactivate the trypsin, FBS was used in a 1:1 ratio, and the whole suspension was subsequently centrifuged at 400× *g* for 10 min. The supernatant was removed, and the cell population was plated on T75 culture flasks at a concentration of 1.5 × 10^6^ cells/flask.

### 2.5. Expression of Surface Markers

Samples were analysed for mesenchymal stromal cell markers (i.e., CD29, CD44, CD90 and CD105) and for hematopoietic stem cells marker (i.e., CD45). Each sample was diluted to a final concentration of 2 × 10^5^ cells/mL and centrifuged at 400× *g*/5 min. Subsequently, the supernatant was removed and the cell pellet was resuspended in 100 μL of PBS containing 3–5 μL of CD90 ((YKIX337.217, allophycocyanin; APC), CD29 ((MEM-101, phycoerythrin; PE), CD44 (MEM-263, APC), CD105 ((MA1-19594, fluorescein isothiocyanate; FITC) and CD45 (YKIX716.13, PE)—all ThermoFisher, Waltham, MA, USA—and incubated for 60 min at 4 °C in the dark. At the end of the incubation period, the samples were centrifuged again at 400× *g*/5 min, the supernatant was removed and the sample was washed in 200–500 µL of washing solution (1% FBS in PBS + 0.1% Sodium Azide (SevernBiotech Ltd., Kidderminster, UK). Cytometric analysis was performed on a BD FACSCanto^®^ flow cytometer (Becton Dickinson Biosciences, San Jose, CA, USA) equipped with a blue (488 nm) and red (633 nm) laser and 6 fluorescence detectors. The percentage of cells expressing individual CD traits was determined by a histogram for the respective fluorescence. The data obtained via measurement were analysed in BD FACS Diva^TM^ analysis software. As a negative control, we used the same type of non-labelled MSCs for autofluorescence control. A gating strategy for flow cytometry was performed by forward/sideward scatter and sideward scatter/sideward scatter pulse height for the elimination of debris and doublets. 

### 2.6. Multilineage Potential

To confirm the multilineage potential of the MSCs, we used the StemProMultilineage differential Kit (Gibco) according to the recommended protocol attached. The cells used for multilineage differentiation were from passage 3 (P3). Cells were cultured in 24-well plates with an initial density of 4 × 10^4^ cells/well for osteocytes, 8 × 10^4^ cells/well for adipocytes and 8 × 10^4^ cells per micromass/well for chondrocytes. Each micromass was a single drop of 5 μL/8 × 10^4^ cells, which was placed in the centre of the well and then incubated at 37 °C and 5% CO_2_ for 2 h for better adherence to the surface, and then 500 μL chondrogenic medium was added. After the recommended culture time (21 days, day in vitro; DIV 21), the cells were fixed using 4% paraformaldehyde (PFA), and the individual populations were stained with the Alizarin red staining method (Sigma) for evidence of calcium deposits in the osteoblast population; Alcian blue (Sigma) for the detection of proteoglycans in the chondroblast population and Oil red (Sigma) for the staining of fat vacuoles in the adipocyte population.

### 2.7. Proliferation Activity of MSC

For the description of cell proliferation, we used the MTT cell proliferation assay kit (Invitrogen). The cells from each population in P3 were plated in 96-well plates, seeding density 1 × 10^4^ cell/well and cultivated in standard cultivation medium (DMEM-F12 w/o phenol red + 10% FBS + 2% ATB + ATM) at 37 °C and 5% CO_2_ for 24, 48, 168 and 240 h. After the cultivation period, we removed the medium and replaced it with 100 µL of fresh culture medium. In the next step, 10 µL of 12 mM MTT stock solution to each well was added and incubated at 37 °C for 4 h. At the end of incubation time, we added 100 µL of SDS-HCl solution, mixed it and incubated. After 12 h, the content of each well was mixed carefully by the pipette and the absorbance was measured at 572 nm by Perkin Elmer Victor3 Multilabel Plate Reader. Statistical analyses were processed via two-way ANOVA, followed by the Tukey test, with the mean considered from five measurements.

### 2.8. Freezing Protocol

At the end of study, MSCs were detached from the culture using 0.25% Trypsin EDTA, washed in FBS and suspended in a prechilled freezing medium consisting of 10% dimetylsulfoxid (DMSO; CryoSure, Wak-Chemie Medical GmbH) + 40% FBS and 50% DMEM-F12. Cells were dispended in 1.8 mL cryovials tubes in concentration 1 × 10^6^ cells/mL. Cryovials were slowly cooled in a Mr. Frosty freezing container and placed in −80 °C freezer before transferring to the vapor of liquid nitrogen at ≤−140 °C. 

## 3. Results

### 3.1. Isolation of Canine MSCs from Bone Marrow, Adipose Tissue and Amnion

Using the abovementioned protocol, we were able to isolate and cultivate a homogeneous population of canine MSCs from bone marrow, adipose tissue and amnion. The yield of isolated cells varied between 1 and 7 × 10^6^ cells/mL (BM-MSC—1 × 10^3^ cells/mL of bone marrow aspirate, AT-MSC—2.5 × 10^6^ cells/g of adipose tissue and AM-MSC—5.6 × 10^6^ cells/g of amniotic tissue) shown in Table 1. Bone marrow and amniotic mesenchymal stem cells showed a fibroblast-like shape (Figure 1A,C and Figure 2(A1,A2,C1,C2)), which is typical for MSCs, in contrast to adipose tissue mesenchymal stem cells, which were longer (approximately 20 µm) and thinner, revealing a spindle-shaped morphology (Figure 1B and Figure 2(B1,B2)).

### 3.2. CD Characterization of Canine MSC

Results of CD analyses (Figure 3) show, that passaging is a suitable tool for obtaining high homogeneity and uniformity of the population during cultivating mesenchymal cells, even in the low passage (all results from passage 3). BM-MSC showed high expression of CD29 (98.7% ± 1.5%), CD44 (97.3% ± 1.0%), CD90 (76.3% ± 2.6%) and CD105 (99.8% ± 3.1%), but low expression of CD45 (3.3% ± 0.3%) on the other hand. AT-MSC were positive for CD29 (99.3% ± 0.7%), CD44 (99.1% ± 0.5%), CD90 (85.9% ± 0.6%), CD105 (99.7% ± 1.2%) and negative for CD45 (1.0% ± 0.2%). Results of different phenotype, AM-MSC, showed positivity for CD29 (72.3% ± 0.8%), CD44 (71.5% ± 1.6%), CD90 (5.8% ± 0.9%), CD105 (90.5% ± 2.9%) and CD45 (3.1% ± 0.4%). We also observed differences in the autofluorescence. The BM-MSCs showed minimal auto autofluorescent florescent positivity for PE (1.1% ± 0.1%), FITC (1.2% ± 0.05%) and APC (2.5% ± 0.5%). The AT-MSCs showed a high number of autofluorescent cells for APC (4.0% ± 0.2%) and a minimum of PE-autofluorescent cells (0.3% ± 0.05%) and FITC (0.6% ± 0.01%). We observed the highest number of autofluorescent cells in AM-MSCs: PE (1.8% ± 0.3%), APC (4.5% ± 0.9%) and FITC (2.9% ± 0.5%). The gating strategy—forward/sideward scatter and sideward scatter/sideward scatter pulse height for the elimination of debris and doublets (Figure 4.). The viability of the observed cells varied between 85 and 93%.

### 3.3. Multilineage Potential

Using a multilineage differentiation kit and the recommended culture protocol and staining methods, we confirmed the ability of MSCs isolated from canine bone marrow, adipose tissue and amnion to differentiate into osteocytes, chondrocytes and only a weak ability to differentiate into adipocytes (Figure 5). All types of isolated canine MSCs (BM-MSCs, AT-MSCs and AM-MSCs) showed high osteogenic and chondrogenic potential; however, the adipogenic differentiation capacity was limited.

### 3.4. MTT Assay

The MTT assay showed that MSCs isolated from all sources have good proliferation capacity; however, we could observe differences in cultivation periods and types of MSCs as well. The best proliferation capacity in vitro was observed in AT-MSCs and at 168 h. The lowest capacity was shown in AM-MSCs. All types of MSCs reached the maximum capacity at 168 h, after this time period, the proliferation capacity decreased (Figure 6).

We also observed differences when comparing the growth rate of cell populations from individual sources after isolation. We achieved the fastest confluent population in adipose tissue-derived MSCs (day 11 of culture). On the contrary, the slowest growth was seen in the bone marrow in which we reached a confluent population on the 29th day of cultivation.

## 4. Discussion

The main goal of this study was to compare the yield, morphology, phenotype, multilineage potential, and proliferation activity of mesenchymal stem cells isolated from different canine tissues—bone marrow, adipose tissue and amniotic tissue. We also met the criteria of the International Stem Cell Research Society (ISSCR) [6,19].

For isolation of MSCs from bone marrow, we used a simple method of centrifugation and size separation using a cell strainer (100 µm). To isolate MSCs from adipose tissue and amnion, we used a combined method of mechanical disruption and enzymatic digestion. As previously described, prolonged digestion can damage the cells [25]. Therefore, we optimised the enzymatic process within the range 25–45 min, depending on the amount and size of the digested fraction at a temperature of 37 °C. For the comparison of isolation yield, the AM-MSCs showed the highest number of isolated cells (up to 5.6 × 10^6^ cells/g). However, the isolation yield of BM-MSCs and AT-MSCs was also not negligible (1 × 10^3^ cells/mL BM-MSC and 2.5 × 10^6^ cells/g AT-MSC). This observation was also confirmed by other studies [26,27,28,29,30]. Concluding from the abovementioned facts, amnion tissue represents a rich source of stem cells. Its collection and retrieval did not cause any added stress and suffering for the animal, and since it had no further use in normal veterinary practice, it would end up as biological waste. Therefore, this tissue source certainly deserves attention in the field of regenerative medicine.

Differences were observed in morphology, as well. While BM-MSCs were characterised by a fibroblast-like shape and a size of 100–120 μm, MSCs isolated from adipose tissue were longer and thinner (100–140 μm), and MSCs isolated from amnion tissue were similar to BM-MSCs (with a fibroblast-like shape) but at first sight morphologically different (oval-shaped) and smaller (80 μm). It was interesting to discover the multilineage differentiation ability of isolated cell populations. After reaching confluence in the P3 and subsequent trypsinization, we tested the cells from each population to confirm the ability to differentiate into osteogenic, chondrogenic, and adipogenic lines. MSCs isolated from canine bone marrow, adipose tissue and amnion showed a very good ability to differentiate into osteogenic and chondrogenic lines but repeatedly very little or lacked the ability to differentiate into adipogenic lines. This variation in the inability of MSCs to differentiate into an adipogenic line has also been described in other studies [31,32]. This may be because cells isolated from the same tissue type but from different collection sites sometimes need up to twice the time to differentiate. For example, the differentiation capacity of bone-marrow-derived MSCs may require a longer time for adipogenic differentiation in comparison to adipose-derived MSCs or amniotic-derived MSCs. Therefore, this has to be optimised for each cell type processed for differentiation [16]. In addition, osteogenesis-mediating and adipogenesis-suppressing bioactive molecules, such as Runx2, Wnt10b and RhoA, or adipogenesis-enhancing bioactive molecules suppressing osteogenesis, including PPARγ, P2X6, LIF, sFRP-1 and BMPs, also play an important role [33]. Thus, osteogenic and chondrogenic differentiation are among the key properties required for hard tissue regeneration in veterinary medicine. This has been confirmed in our study in all MSC types, which nominates them as candidates for possible treatment application [34]. The expression of CD surface markers in MSC populations was tested using flow cytometry. During the analysis, we found a lower percentage of CD29+, CD44+ and CD105+ and almost no positivity for CD90 in AM-MSCs compared with BM-MSCs and AT-MCs, where the expression of all four CD markers was high [35]. The majority of canine papers have demonstrated either a single alternative MSC marker (CD29 or CD44) or both, which is more consistent than any of the classic MSC markers [15,23,24,36]. High expression of CD29 and CD105 is superior to strong chondrogenic potential by regulating TGF-β/Smad2 [37]. The same can be said about the expression of CD44 describing chondrogenic potential via the Smad and ERK signalling pathways [38]. All of these facts correlate to our results of chondrogenic differentiation. Both CD29 and CD44 expression were found to be involved with MSC adhesion, migration and engraftment in vivo [15]. We were able to confirm low or no adipogenic potential in the studied MSCs. This could be caused by high number of CD29 positive cells in the observed population, because CD29 and CD90 reduce adipogenic capacity [39]. The absence of or low positivity for CD90, in the case of canine AM-MSCs, may be associated with their low immunosuppressive properties; similarly, it has been reported in human MSCs [40]. Another study referred to the loss of CD90 (THY-1) as to a process that may increase MSC differentiation potential, which is often associated with a decrease in CD44 expression [41]. Thus, although AM-MSCs are a rich source with good differentiation potential, we have to keep in mind possible differences due to the difference in their neonatal origin. Finally, it should be noted that the actual expression of surface markers also depends on the MSC isolation source [42] as well as on the donor age and cell passage [43]. Despite a previous study [44], higher expression of CD45 in the observed population of BM-MSCs and AM-MSCs (>3%) had no pronounced impact on multilineage potential compared to AT-MSCs. In addition, obvious differences in morphology, phenotype, and proliferative activity in individual cell populations were also observed in a study led by Bearden [23]. 

The autofluorescence of MSCs at 488 nm correlates with cell granularity and the expression of CD90. Increased autofluorescence was negatively associated with the expression of CD90, which was also confirmed in our study [45]. Furthermore, cells revealing high granularity (i.e., AM-MSCs) showed low expression of CD90 and high autofluorescence. 

In this study, differences in the proliferation capacity of MSCs from different sources were noticed using the MTT assay. The AT-MSCs showed the highest capacity, while AM-MSCs showed a high yield of cells but revealed the lowest proliferation activity. All three types of MSCs reached their proliferation peak between 168 and 240 h of cultivation. After this cultivation period, the proliferation capacity decreased. This fact could correlate with the confluency of the given population and cell density. Similar results were published recently [26,28].

All of the abovementioned facts create preconditions for the use of MSCs isolated from all three tissues in cell therapy in veterinary medicine [11]. However, it should be emphasised that each MSC originating from different tissues should be thoroughly characterised and, afterward, carefully considered and adapted to the patient’s condition and nature.

## 5. Conclusions

In this study, we described the effective protocols for the isolation of MSCs from different canine tissues—bone marrow, adipose tissue and amnion membrane. Our results presented differences in yield of isolation, morphology, phenotype, multilineage potential and proliferation activity. All types of isolated canine MSCs expressed CD105 surface markers, but not CD45, which correlated to their high chondrogenic capacity. The AT-MSCs showed the best phenotype homogeneity, proliferation capacity, multilineage potential and even high yield of isolation (Table 1). All the presented data create the foundation for their further investigation and potential use in regenerative therapy.

## Figures and Tables

**Figure 1 animals-12-01502-f001:**
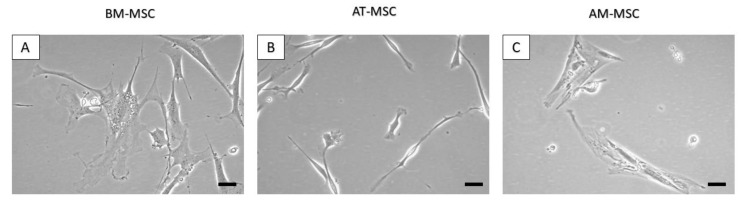
Detailed morphology of canine MSCs at passage 0 (P0) day in vitro (DIV) 3 from different sources. The BM-MSCs (**A**) and AM-MSCs (**C**) revealed a fibroblast-like morphology, while the AT-MSCs (**B**) showed a spindle-like morphology. Scale bars = 20 µm.

**Figure 2 animals-12-01502-f002:**
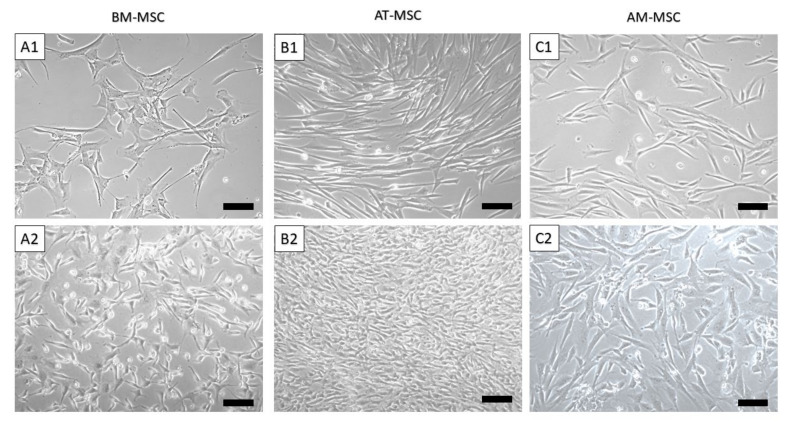
Morphology of canine MSCs from different sources. BM-MSCs at passage 0 (P0) day in vitro (DIV) 5 (**A1**) and DIV15 (**A2**); AT-MSCs at passage P0 DIV5 (**B1**) and DIV15 (**B2**); AM-MSCs at passage P0 DIV5 (**C1**) and DIV15 (**C2**). Scale bars = 50 µm.

**Figure 3 animals-12-01502-f003:**
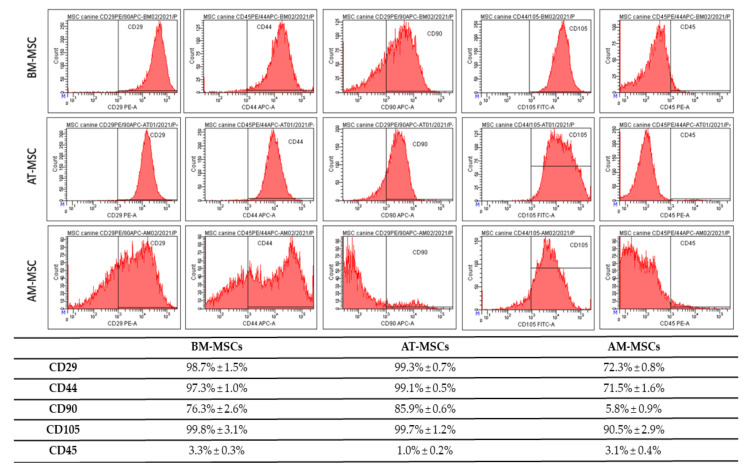
Results of the CD analyses from passage 3 (P3). The BM-MSCs showed positivity for CD29 (98.7% ± 1.5%), CD44 (97.3% ± 1.0%), CD90 (76.3% ± 2.6%), CD105 (99.8% ± 3.1%) and low expression for CD45 (3.3% ± 0.3%). The AT-MSCs showed high expression of CD29 (99.3% ± 0.7%), CD44 (99.1% ± 0.5%), CD90 (85.9% ± 0.6%), CD105 (99.7% ± 1.2%) and low expression of CD45 (1.0% ± 0.2%). The AM-MSCs showed positivity for CD29 (72.3% ± 0.8%), CD44 (71.5% ± 1.6%), CD90 (5.8% ± 0.9%), CD105 (90.5% ± 2.9%) and low expression of CD45 (3.1% ± 0.4%).

**Figure 4 animals-12-01502-f004:**
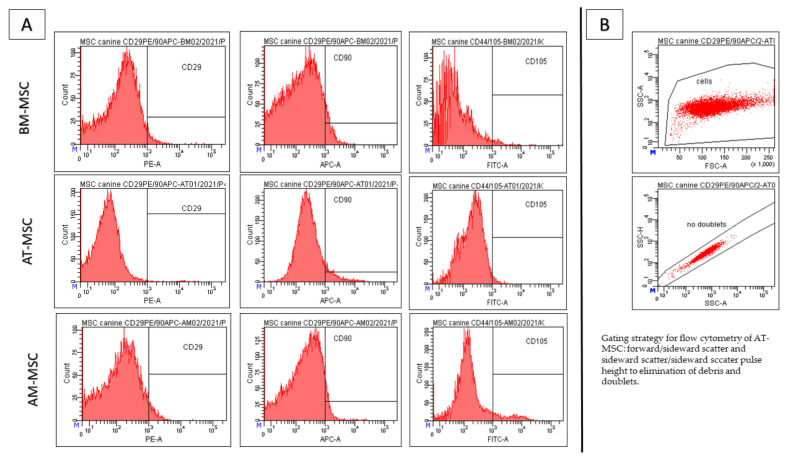
Results of the CD analyses from passage 3 (P3)—negative control (**A**) and Gating strategy (**B**). The BM-MSCs showed minimal autofluorescent positivity for PE (1.1% ± 0.1%), FITC (1.2% ± 0.05%) and APC (2.5% ± 0.5%). The AT-MSCs showed a high number of autofluorescent cells for APC (4.0% ± 0.2%) and a minimum of PE-autofluorescent cells (0.3% ± 0.0 5%) and FITC (0.6% ± 0.01%). We observed the highest number of autofluorescent cells in AM-MSCs: PE (1.8% ± 0.3%), APC (4.5% ± 0.9%) and FITC (2.9% ± 0.5%). Gating strategy—forward/sideward scatter and sideward scatter/sideward scatter pulse height for the elimination of debris and doublets (Figure 4).

**Figure 5 animals-12-01502-f005:**
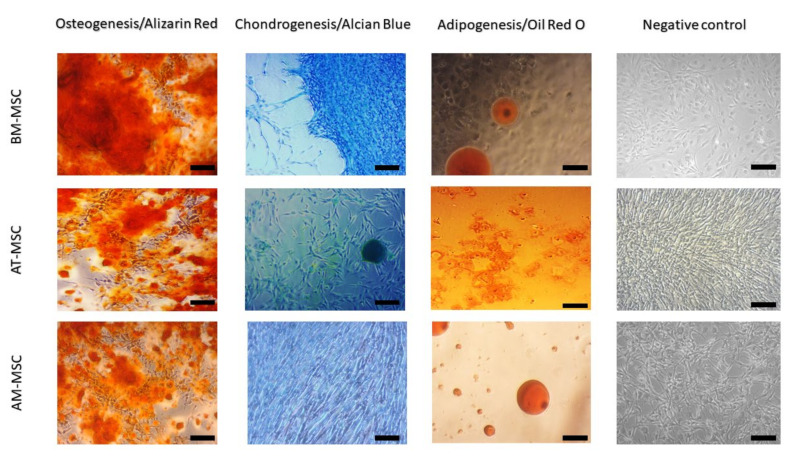
Multilineage potential of canine MSCs. All types of canine MSCs showed high osteogenic (presence of calcium deposits detected by Alizarin red) and chondrogenic potential (presence of glycoproteoglycanes detected by Alcian blue staining); however, there was the absence (BM-MSCs and AM-MSCs) or low (AT-MSCs) level of the adipogenic potential (triglycerides detected by Oil Red O staining). Scale bars = 50 µm.

**Figure 6 animals-12-01502-f006:**
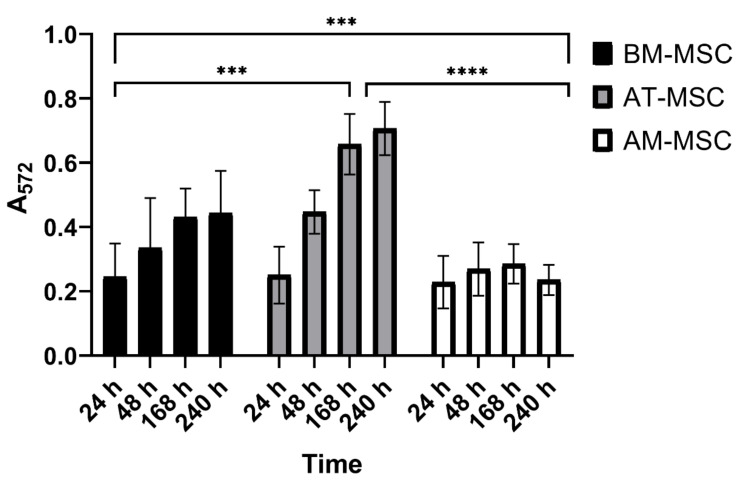
Results of the MTT assay. Graph representing absorbance measured after 24, 48, 168, and 240 h at 572 nm, which correlates with the proliferation capacity of examined types of MSCs. *** *p* = 0.0007 and **** *p* < 0.0001.

**Table 1 animals-12-01502-t001:** Comparative analysis of the common MSC properties from canine MSCs isolated from bone marrow, adipose tissue and amnion. +++, ++, + and – indicate high, moderate, low or absence of mentioned properties, respectively.

Results of the Comparative Study of Canine MSCs			
	BM-MSCs	AT-MSCs	AM-MSCs
Invasiveness of tissue collection	+++	+++	–
Yield	+	++	+++
Homogeneity	+	+++	+
Osteogenic potential	+++	+++	+++
Chondrogenic potential	+++	+++	+++
Adipogenic potential	–	+	–
Proliferation capacity	++	+++	+

## Data Availability

The data presented in this study are available upon request from the corresponding author.

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
