# Peer review of "A Comparative Study of Canine Mesenchymal Stem Cells Isolated from Different Sources"

_animals, 2022, doi:10.3390/ani12121502_

Round 1
Reviewer 1 Report
There are some limitations in explaining the terms used in the articles which are already mentioned below. Along with this since poor English and terminology are used which makes the manuscript even more complicated that’s why it needs a full proofreading to bring clarity in the meaning of how the experiments had been carried out. Although the experimental analysis is good enough but since in discussing and the introduction part needs some little more work to do. In the article the at some points the journals guidelines were not properly followed.
L1-2: The title of this article should be change to Comparative Study of Canine Mesenchymal Stem Cells isolated from Different Sources.
L22: The abstract portions need proper proofreading, and the terminology should be address properly.
L24-25: Many studies have been conducted to develop alternative products with enhanced efficacy and safety. (Provide the reference of the studies that are done on developing alternative products.)
L234-236: After reaching confluence in the third passage (P3) and subsequent trypsinization, we tested the cells from each population to confirm the ability to differentiate into an osteogenic, chondrogenic, and adipogenic line. (Provide the result for confirmation)
L250-251: This has been confirmed in our study in all 250 MSCs types, which nominates them as candidates for possible treatment application. (Provide reference or data to support this statement).
Author Response
Revision 1
Dear Reviewer 1,
Please find enclosed a revised manuscript with point-by-point responses to your comments. We are thankful for your valuable and constructive comments and suggestions. We appreciate the time and effort that you dedicated to providing valuable feedback on our manuscript. We have incorporated the changes to reflect most of your suggestions. Modifications made in the revised manuscript are marked up using the “Track Changes” on.
Reviewer 1
There are some limitations in explaining the terms used in the articles which are already mentioned below. Along with this since poor English and terminology are used which makes the manuscript even more complicated that’s why it needs a full proofreading to bring clarity in the meaning of how the experiments had been carried out. Although the experimental analysis is good enough but since in discussing and the introduction part needs some little more work to do. In the article the at some points the journals guidelines were not properly followed.
L1-2: The title of this article should be change to Comparative Study of Canine Mesenchymal Stem Cells isolated from Different Sources.
Response: The title of our manuscript was changed according to your suggestion
L22: The abstract portions need proper proofreading, and the terminology should be address properly.
Response: The abstract was modified.
L24-25: Many studies have been conducted to develop alternative products with enhanced efficacy and safety. (Provide the reference of the studies that are done on developing alternative products.)
Response: Reference was provided L73 (Sasaki, A.; Mizuno, M.; Ozeki, N.; Katano, H.; Otabe, K.; Tsuji, K.; Koga, H.; Mochizuki, M.; Sekiya, I. Canine Mesenchymal Stem Cells from Synovium Have a Higher Chondrogenic Potential than Those from Infrapatellar Fat Pad, Adipose Tissue, and Bone Marrow. PloS One 2018, 13, e0202922, doi:10.1371/journal.pone.0202922.)
L234-236: After reaching confluence in the third passage (P3) and subsequent trypsinization, we tested the cells from each population to confirm the ability to differentiate into an osteogenic, chondrogenic, and adipogenic line. (Provide the result for confirmation)
Response: Confirmation was provided in Results, please see lines L250-L252
L250-251: This has been confirmed in our study in all 250 MSCs types, which nominates them as candidates for possible treatment application. (Provide reference or data to support this statement).
Response: We have provided a reference (Miao, C.; Lei, M.; Hu, W.; Han, S.; Wang, Q. A Brief Review: The Therapeutic Potential of Bone Marrow Mesenchymal Stem Cells in Myocardial Infarction. Stem Cell Res. Ther. 2017, 8, 242, doi:10.1186/s13287-017-0697-9.). Please see line L353.

Reviewer 2 Report
The manuscript compared three different types of canine MSCs.
But some points need to be improved.
Introduction
There are other papers published comparing different types of canine MSCs and several about canine Adipose-derived stem cells isolated and characterized. These need to be citated in the introductions and in the Discussion sections.
Revise the title, please.
The Animal ethical committee need to be presented.
Materials and Methods
The concentration of each reagent needs to be presented and not the volume.
Include more information about the donor animals: sex, age, weight. Are they healthy dogs?
Inform the code, clone, reactivity, specificity of each antibody
Cit the N of each experiment including the number of replicates and independent experiments.
Describe the freezing protocol of the cells.
Cytometry:
What was the blocking and wash solution used? What was the percentage of live cells?
It is necessary to show the gate of the cells and negative control.
According ISSCR a panel of markers was defined to characterize MSC cells. Although the authors used some of them, other antibodies need to be included in this comparative study to characterize well their cells.
It is necessary to show negative markers for these cells and others like CD73 or CD105. Cd34 is essential to define the kind of cells.
Minor point:
Define DIV in the text
There are some syntaxes errors- the name of reagents
Results
What are the black dots into the Figures A1-C1? The cells not seem clean.
Author Response
Revision 1
Dear Reviewer 2,
Please find enclosed a revised manuscript with point-by-point responses to your comments. We are thankful for your valuable and constructive comments and suggestions. We appreciate the time and effort that you dedicated to providing valuable feedback on our manuscript. We have incorporated the changes to reflect most of your suggestions. Modifications made in the revised manuscript are marked up using the “Track Changes” on.
Reviewer 2
Comments and Suggestions for Authors
The manuscript compared three different types of canine MSCs.
But some points need to be improved.
Introduction
There are other papers published comparing different types of canine MSCs and several about canine Adipose-derived stem cells isolated and characterized. These need to be citated in the introductions and in the Discussion sections.
Response: Canine papers are cited in the introduction and in the discussion, too. (Rashid, U.; Yousaf, A.; Yaqoob, M.; Saba, E.; Moaeen-ud-Din, M.; Waseem, S.; Becker, S.K.; Sponder, G.; Aschenbach, J.R.; Sandhu, M.A. Characterization and Differentiation Potential of Mesenchymal Stem Cells Isolated from Multiple Canine Adipose Tissue Sources. BMC Vet. Res. 2021, 17, 388, doi:10.1186/s12917-021-03100-8.); (Bearden, R.N.; Huggins, S.S.; Cummings, K.J.; Smith, R.; Gregory, C.A.; Saunders, W.B. In-Vitro Characterization of Canine Multipotent Stromal Cells Isolated from Synovium, Bone Marrow, and Adipose Tissue: A Donor-Matched Comparative Study. Stem Cell Res. Ther. 2017, 8, 218, doi:10.1186/s13287-017-0639-6.); (Kisiel, A.H.; McDuffee, L.A.; Masaoud, E.; Bailey, T.R.; Esparza Gonzalez, B.P.; Nino-Fong, R. Isolation, Characterization, and in Vitro Proliferation of Canine Mesenchymal Stem Cells Derived from Bone Marrow, Adipose Tissue, Muscle, and Periosteum. Am. J. Vet. Res. 2012, 73, 1305–1317, doi:10.2460/ajvr.73.8.1305.); (Bearden, R.N.; Huggins, S.S.; Cummings, K.J.; Smith, R.; Gregory, C.A.; Saunders, W.B. In-Vitro Characterization of Canine Multipotent Stromal Cells Isolated from Synovium, Bone Marrow, and Adipose Tissue: A Donor-Matched Comparative Study. Stem Cell Res. Ther. 2017, 8, 218, doi:10.1186/s13287-017-0639-6.)
Revise the title, please.
Response: The title was changed to: A Comparative Study of Canine Mesenchymal Stem Cells Isolated from Different Sources, please see lines L1-L2
The Animal ethical committee need to be presented.
Response: The approval of the Ethical comittee was completed (EKVP / 2021-01), please see line L82
Materials and Methods
The concentration of each reagent needs to be presented and not the volume.
Response: The concentrations of reagents are described in the text (enzymes, antibiotics, glutamine, FBS), L95-L97, L111-L114,
Include more information about the donor animals: sex, age, weight. Are they healthy dogs?
Response: Information about donors was completed according to requirements. Please see lines L85-L87, L117-L119, L120-L127
Inform the code, clone, reactivity, specificity of each antibody
Response: Specification of each antibody was added L142-L145 (code, clonality and dyes)
Cit the N of each experiment including the number of replicates and independent experiments.
Response: N is citated in each paragraph (2.1.; 2.2.; 2.3.) of methodology, please see L85, L102 and L117
Describe the freezing protocol of the cells.
Response: Freezing protocol is described as section 2.8. Freezing protocol. L186-L192
Cytometry:
What was the blocking and wash solution used? What was the percentage of live cells?
Response: The data were completed – composition of wash solution and viability is described in text. Please see lines L148, L156-L157
It is necessary to show the gate of the cells and negative control.
Response: Gating strategy is described in the methodology and in the Figure 4. together with negative controls, please see in the text Lines 154-157.
According ISSCR a panel of markers was defined to characterize MSC cells. Although the authors used some of them, other antibodies need to be included in this comparative study to characterize well their cells.
It is necessary to show negative markers for these cells and others like CD73 or CD105. Cd34 is essential to define the kind of cells.
Response: From proposed markers we chose CD105. Please see lines L215-231. We were not able to complete analyse for CD34, because of lack CD34 antibody. Similarly, other authors didn´t examined MSC for CD34 (Hendawy et al. 2021, Tissue harvesting side effect on the canine adipose stromal vascular fraction quantity and quality, in Animals); (Yanesselli et al. 2013, Allogenic stem cell transplantation for bone regeneration of non union defect in a canine, in Veterinary Medicine: Research and reports;) (Rashid et al. 2021, Characterization and differentiation potential of mesenchymal stem cells isolated from multiple canine tissue sources)
Minor point:
Define DIV in the text
Response: DIV was defined in the text L167
There are some syntaxes errors- the name of reagents
Response: Errors were checked and corrected.
Results
What are the black dots into the Figures A1-C1? The cells not seem clean.
Response: Fig. 2. A1-C1 was replaced by a new figure.

Reviewer 3 Report
The authors of this manuscript aimed to compare the morphology, phenotype, multilineage potential and proliferation activity of mesenchymal stem cells isolated from canine bone marrow, adipose tissue and amniotic tissue. The tissue of origin of MSCs is often investigated and a comparative study between the different tissues could provide important information for veterinary regenerative medicine; however important improvements have to be made to the manuscript.
Major issues
The introduction needs to be revised. The study should be placed in a broader context, and it should be emphasized why it is important in the field of canine regenerative medicine. The purpose of the study and its significance should be also defined, highlighting controversial and divergent hypotheses when necessary and citing key publications. Finally, the introduction should briefly mention the primary aim of the work and highlight the most important conclusions.
The materials and methods should be improved. It is not clear why the authors specify that the dogs used are purebred: Was informed consent required to dog-owners? Moreover, an ethic statement is not reported in the manuscript.
The results are incomplete, they do not report some data that are discussed in the discussion section. Table 1 is not cited and is not described in the text.
The discussions must be revised as a result of the modification of the rest of the text of the manuscript.
Minor issues
Line 36: replace "specialized" with "unspecialized"
From line 57 check all the text of the manuscript and replace "ml" with "mL"
Line 60: the first time that appears in the text "ATB-ATM" write the name in full: antibiotic-antimicotic (ATB-ATM) (penicillin-streptomycin-amphotericin B (Biowest). It is not necessary to repeat it always in the text, after you can report only the acronym.
Line 64; 77; 94: delete the two "(Biowest)".
Line 64-65; 77-78;86; 92; 94-95: delete “(penicillin-streptomycin-amphotericin B) (Biowest)”.
Line 68: rewrite the sentence: "adipose tissue was harvested from purebred large dogs (n = 3) ..."
Line 70: replace “fatty” with “adipose”.
Line 71: is the "phosphate buffer" the Phosphate-Buffered Saline? If yes, write here for the first time "Phosphate-Buffered Saline (PBS, Biowest)" appears; subsequently, only the acronym PBS can be reported in the text (lines 85, 110, 114).
Line 78: rewrite the sentence "... and plated in a T25 tissue culture flask at a concentration ...."
Lines 93 and 104: delete “(TPP)”.
Line 93: the cell concentration is 106 cells/mL or 106 cells/ flask?
Line 124: replace "40,000 cells" with "4x104 cells".
Lines 124,126: replace "80,000 cells" with "8x104 cells".
Line 137: replace “0,01x106 cells” with “104 cells".
Line 144: which instrumentation was used for absorbance measurement?
Line 174: delete "(Gibco)".
Lines 204-205: the authors report that they have standardized protocols for the isolation of MSCs but it is not clear how they have been standardized.
The authors report having observed a different morphology between the cells, but this is not appreciated from the images and little from the description made in paragraph 3.1. Spindle cells are cells that are longer than they are wide. The same is true for fibroblast-like shaped cells. This aspect should be better exposed.
The results lack information on cell yield from various tissue types and other information which is discussed in the discussion section.
Discussions should be revised based on changes made in the rest of the manuscript text. In particular, the authors should better deepen the discussion concerning the results obtained for the expression of CDs of canine MSCs.
Author Response
Revision 1
Reviewer 3
Dear Reviewer 3,
Please find enclosed a revised manuscript with point-by-point responses to your comments. We are thankful for your valuable and constructive comments and suggestions. We appreciate the time and effort that you dedicated to providing valuable feedback on our manuscript. We have incorporated the changes to reflect most of your suggestions. Modifications made in the revised manuscript are marked up using the “Track Changes” on.
Comments and Suggestions for Authors
The authors of this manuscript aimed to compare the morphology, phenotype, multilineage potential and proliferation activity of mesenchymal stem cells isolated from canine bone marrow, adipose tissue and amniotic tissue. The tissue of origin of MSCs is often investigated and a comparative study between the different tissues could provide important information for veterinary regenerative medicine; however important improvements have to be made to the manuscript.
Major issues
The introduction needs to be revised. The study should be placed in a broader context, and it should be emphasized why it is important in the field of canine regenerative medicine. The purpose of the study and its significance should be also defined, highlighting controversial and divergent hypotheses when necessary and citing key publications. Finally, the introduction should briefly mention the primary aim of the work and highlight the most important conclusions.
Response: The introduction has been significantly modified in light of the opponent's comments.
The materials and methods should be improved. It is not clear why the authors specify that the dogs used are purebred: Was informed consent required to dog-owners? Moreover, an ethic statement is not reported in the manuscript.
Response: Information about donors was completed (breed, sex, age, weight). The number fo Ethical committee approval was added.
Please see lines L81-L82 L85-L87, L116-L119.
The results are incomplete, they do not report some data that are discussed in the discussion section. Table 1 is not cited and is not described in the text.
Response: All results are discussed in Discussion; Tab. 1 is mentioned in results and in the conclusion. Please see lines L201, L363.
The discussions must be revised as a result of the modification of the rest of the text of the manuscript.
Response: Discussion was modified and results data were added.
Minor issues
Line 36: replace "specialized" with "unspecialized"
From line 57 check all the text of the manuscript and replace "ml" with "mL"
Line 60: the first time that appears in the text "ATB-ATM" write the name in full: antibiotic-antimicotic (ATB-ATM) (penicillin-streptomycin-amphotericin B (Biowest). It is not necessary to repeat it always in the text, after you can report only the acronym.
Line 64; 77; 94: delete the two "(Biowest)".
Line 64-65; 77-78;86; 92; 94-95: delete “(penicillin-streptomycin-amphotericin B) (Biowest)”.
Line 68: rewrite the sentence: "adipose tissue was harvested from purebred large dogs (n = 3) ..."
Line 70: replace “fatty” with “adipose”.
Line 71: is the "phosphate buffer" the Phosphate-Buffered Saline? If yes, write here for the first time "Phosphate-Buffered Saline (PBS, Biowest)" appears; subsequently, only the acronym PBS can be reported in the text (lines 85, 110, 114).
Line 78: rewrite the sentence "... and plated in a T25 tissue culture flask at a concentration ...."
Lines 93 and 104: delete “(TPP)”.
Line 93: the cell concentration is 106 cells/mL or 106 cells/ flask?
Line 124: replace "40,000 cells" with "4x104 cells".
Lines 124,126: replace "80,000 cells" with "8x104 cells".
Line 137: replace “0,01x106 cells” with “104 cells".
Response: All issues were changed and corrected according to suggestions.
Line 144: which instrumentation was used for absorbance measurement?
Response: Perkin Elmer Victor3 Multilabel Plate Reader was used for absorbance measurements – Name of instrument added to methodology. Please see line L182.
Line 174: delete "(Gibco)".
Lines 204-205: the authors report that they have standardized protocols for the isolation of MSCs but it is not clear how they have been standardized.
Response: All issues were changed according suggestions.
The authors report having observed a different morphology between the cells, but this is not appreciated from the images and little from the description made in paragraph 3.1. Spindle cells are cells that are longer than they are wide. The same is true for fibroblast-like shaped cells. This aspect should be better exposed.
Response: Fig.1 was added, where the details of morphology was described. Please see lines L197-L205.
The results lack information on cell yield from various tissue types and other information which is discussed in the discussion section.
Response: Cells yields from each tissue is described in part Results – 3.1 Isolation of canine MSC from bone marrow, adipose tissue and amnion. Please see lines L198-L201.
Discussions should be revised based on changes made in the rest of the manuscript text. In particular, the authors should better deepen the discussion concerning the results obtained for the expression of CDs of canine MSCs.
Response: Discussion was revised according the requirements. Please see lines L318-L343.

Round 2
Reviewer 3 Report
The authors have made numerous improvements to the manuscript as suggested. However, some aspects to be revised remain in the text.
In the materials and methods section, the companies listed in brackets after the material used should only be reported the first time the material used appears in the text.
In the section 2.5, the cell concentration is 2x105 / mL?
Lines 209-210, section 2.5, write: "the cell pellet was resuspended in 100uL of The following monoclonal antibodies: CD90 ....." and delete "monoclonal antibodies" inside all parentheses of the sentence.
Lines 226-227: the last sentence of paragraph 2.6 is a result.
The figures reported should be mentioned in the results (figures 1A and C, figure 2A and C)
Table 1 should be reported next to the results.
Author Response
Revision 2
Reviewer 3
Dear Reviewer 3,
Please find enclosed a revised manuscript with point-by-point responses to your comments. We are thankful for your valuable and constructive comments and suggestions. We appreciate the time and effort that you dedicated to providing valuable feedback on our manuscript. We have incorporated the changes to reflect most of your suggestions. Modifications made in the revised manuscript are marked up using the “Track Changes” on.
The authors have made numerous improvements to the manuscript as suggested. However, some aspects to be revised remain in the text.
In the materials and methods section, the companies listed in brackets after the material used should only be reported the first time the material used appears in the text.
Response: We made all changes about companies according your request. Please check the manuscript, part Methodology.
In the section 2.5, the cell concentration is 2x105 / mL?
Response: Volume unit (mL) was completed. Please see lines L-140.
Lines 209-210, section 2.5, write: "the cell pellet was resuspended in 100uL of The following monoclonal antibodies: CD90 ....." and delete "monoclonal antibodies" inside all parentheses of the sentence.
Response: All changes were made up according request. Please see lines L141-144.
Lines 226-227: the last sentence of paragraph 2.6 is a result.
Response: Sentence about viability of cells was transferred to the part Results; Part 3.2. CD characterization of canine MSC, Please see line L 230-231.
The figures reported should be mentioned in the results (figures 1A and C, figure 2A and C)
Response: Reported figures were completed in Results, Part 3.1. Please see line L200-201.
Table 1 should be reported next to the results.
Response: Table 1 was moved behind paragraph 3.4. MTT assay as a sum of all our results.
